# Hedgehog Signaling in CNS Remyelination

**DOI:** 10.3390/cells11142260

**Published:** 2022-07-21

**Authors:** Minxi Fang, Tao Tang, Mengsheng Qiu, Xiaofeng Xu

**Affiliations:** 1Institute of Life Sciences, College of Life and Environmental Sciences, Hangzhou Normal University, Hangzhou 311121, China; mfang3@163.com; 2College of Life Sciences, Zhejiang University, Hangzhou 310058, China; 3Department of Anatomy, Cell Biology & Physiology, School of Medicine, Indiana University, Indianapolis, IN 46202, USA; tangtao@iu.edu; 4School of Basic Medicial Sciences, Hangzhou Normal University, Hangzhou 311121, China

**Keywords:** hedgehog signaling pathway, oligodendrocytes, remyelination, multiple sclerosis

## Abstract

Remyelination is a fundamental repair process in the central nervous system (CNS) that is triggered by demyelinating events. In demyelinating diseases, oligodendrocytes (OLs) are targeted, leading to myelin loss, axonal damage, and severe functional impairment. While spontaneous remyelination often fails in the progression of demyelinating diseases, increased understanding of the mechanisms and identification of targets that regulate myelin regeneration becomes crucial. To date, several signaling pathways have been implicated in the remyelination process, including the Hedgehog (Hh) signaling pathway. This review summarizes the current data concerning the complicated roles of the Hh signaling pathway in the context of remyelination. We will highlight the open issues that have to be clarified prior to bringing molecules targeting the Hh signaling to demyelinating therapy.

## 1. Introduction

### 1.1. Remyelination

Remyelination or myelin regeneration is a fundamental repair process in the central nervous system (CNS) that is triggered by demyelinating events. Multiple sclerosis (MS) is the most common demyelinating disease occurring in the CNS [1]. It is generally thought that demyelination in MS patients is brought about by an autoimmune-mediated attack on the myelin [2]. Spontaneous remyelination occurs at the onset of disease but often fails when the disease progresses, leading to the loss of metabolic support from the myelin to axons with subsequent axon degeneration and, finally, irreversible neurological disabilities. MS is considered the leading cause of non-traumatic disability occurring in young adults. Currently available treatments for MS exclusively target the immune system, relying on multiple immunomodulators to suppress different immune cells or cytokines and reducing inflammatory outbreaks at the early stage of MS [3,4,5,6]. However, inhibiting the recurring inflammation does not necessarily ensure axonal remyelination in the demyelinated lesions. There is currently no treatment available for the progressive stage of MS. Furthermore, the development of remyelinating therapies depends on a better understanding of the mechanisms and targets that regulate myelin regenerative processes.

The bottleneck of remyelination failure is the inability of oligodendrocyte progenitor cells (OPCs) to differentiate into myelinating oligodendrocytes (OLs). In animal models, as well as in humans, myelin regeneration is initiated by both resident OPCs at the site of lesion and neural progenitor cells (NPCs) located in the ventricular-subventricular zone (V-SVZ) of the brain [7,8,9,10,11]. Recently, several lines of evidence from MS patients demonstrated that along with newly generated OPCs, spared OLs during the demyelination process also participate in remyelination [12,13]. The recruitment of these various cell subsets at the site of a lesion and their respective contribution to the remyelination process remain to be clarified.

### 1.2. The Hedgehog (Hh) Signaling Pathway

Several signaling pathways involved in developmental myelination have been implicated in the remyelination program, including the Hh pathway [14]. Molecular and genetic studies have revealed that the Hh signaling pathway constitutes one of the most important pathways that regulate cell fate specification, proliferation, and differentiation during development and tissue regeneration [15,16]. The Hh signaling pathway, which is highly conserved during evolution, was first recognized in *Drosophila*, where it plays an essential role in the segmentation and body-plan patterning of the larva [17]. Three Hh proteins have been identified in vertebrates, including Sonic Hedgehog (Shh), Desert Hedgehog (Dhh), and Indian Hedgehog (Ihh) [18]. Among them, Shh has marked roles during developmental oligodendrogenesis and myelination [19,20,21,22,23]. *Ihh* loss-of-function analysis showed that Ihh is also required for OPC specification from NPCs [24]. In contrast, Dhh primarily regulates the development of the peripheral nerve sheath and is a potential target for treating demyelination in compression neuropathies [25,26].

The activation of Hh signaling can trigger two alternative pathways: canonical and non-canonical (Figure 1). When Hh ligands are absent, The 12-pass transmembrane protein receptor Patched 1 (Ptch1) exerts a repressive activity on the G-protein coupled receptor, Smoothened (Smo). In the canonical pathway, glioma-associated oncogene homolog (Gli) transcription factors are sequestered by suppressors of fused (Sufu) and phosphorylated by PKA, CK1, and GSK-3β, making them for proteolytic cleavage [27]. The cleavage of the C-terminal domain creates GliR, the repressor form of Gli factors, which then translocates into the nuclear and represses downstream Hh target genes (Figure 1A). Upon Hh binding to Ptc1, the repression of Smo is relieved. Smo then accumulates in the primary cilium, inhibits the sequestration and phosphorylation of Gli factors, and promotes their dissociation from Sufu. Full-length Gli factors are transcriptional activators that translocate into the nucleus and activate the transcription of Hh target genes (Figure 1B) [28,29]. Three Gli factors, Gli1, Gli2, and Gli3, are reported to be the primary transcriptional effectors involved in the Hh signaling in vertebrates. They appear to have distinct roles in mediating the Hh signaling of various cell types during development [30,31,32].

In addition to the canonical Hh signaling pathway, a number of pathways triggered by Hh ligands, but outside the Hh-Ptch1-Smo-Gli axis and independent of the primary cilium, have been identified and collectively called ‘non-canonical’ Hh signaling pathways [33,34,35,36], which have been split into two types: Type Ⅰ is Smo-independent, and Type Ⅱ is Smo-dependent. The majority of non-canonical Hh signaling pathways belong to Type Ⅱ. Moreover, the activation of Gli factors that are independent of Hh ligands is also referred to as non-canonical. As a G protein-coupled receptor, Smo can trigger a variety of downstream responses by interacting with the heterotrimer G proteins [37,38]. For instance, it can activate Rac family small GTPase 1 (Rac1) and Ras homologous family member A (RhoA), leading to actin remodeling and subsequent cell migration [39,40]. Src family kinase (SFK) is another effector of Smo that has been shown to regulate axonal growth cones [41]. Smo activation can also trigger a rapid Ca2+ influx, followed by the activation of Calcium/Calmodulin-dependent protein kinase kinase 2 (CaMKK2), which in turn promotes the phosphorylation of AMPK, a sensor of cellular metabolism [42]. Of note, non-canonical Hh signaling can promote ciliogenesis through two distinct pathways. Firstly, Smo directly activates LKB1/AMPK, which results in autophagy and the removal of satellite OFD1, which is an essential step for ciliogenesis. On the other way, the interaction between Smo and heterotrimer G proteins activates the LGN-NuMA-Dynein protein complex, which is important for the delivery of OFD1 to the basal body during ciliogenesis [36]. Since ciliogenesis is required for the canonical Hh signaling pathway, there exists crosstalk between canonical and non-canonical Hh signaling. For detailed information about the Hh signaling, the readers are referred to some comprehensive reviews [33,35,43,44].

### 1.3. Animal Models for Investigation of Demyelination and Remyelination

Since the etiology and histopathology of MS are complex, different types of animal models have been used to study the mechanisms associated with demyelinating diseases and to evaluate the efficacy of novel therapeutic drug candidates. A comparison of histopathological characteristics between different MS animal models is given in [45,46,47]. The most studied MS animal model is the experimental autoimmune encephalomyelitis (EAE), in which inflammation and autoimmune responses are induced by the injection of synthetic peptides derived from myelin proteins. For instance, immunization of SJL/J mice with the immunodominant PLP peptide (PLP_139-151_) induces a relapsing–remitting disease course [48], while the immunization of C57BL6/J mice with MOG_35-55_ peptide induces a chronic course [49]. Even though the EAE model is an excellent tool to study the immune response during different courses of MS, there are also limitations. For example, remyelination is difficult to be studied in an EAE model. Instead, toxic demyelination is more suitable for studying the de- and remyelination process. Cuprizone is a copper chelator, which, when supplementing normal chow, induces oligodendroglial cell death with subsequent demyelination, together with the activation of inflammatory cells. Spontaneous remyelination occurs when removing Cuprizone from the diet. In contrast to the EAE model, demyelination is most pronounced in the corpus callosum and somatosensory cortex region [50]. Similarities between the Cuprizone model and progressive MS pathology are innately-driven myelin and axonal injury, functional activation of oxidative stress pathway [51], and the relative preservation of blood–brain barriers. The microinjection of lysolecithin, which is also termed lysophosphatidyl choline (LPC), into the white matter tracts caused prompt demyelination, followed by remyelination. This model has been used very productively to examine the cellular and molecular determinants of remyelination. Demyelination occurs due to the primary toxic effects of detergent on myelin sheaths rather than the secondary effects on OLs [52]. Moreover, LPC triggers a rapid and high reproducible form of demyelination in the CNS without producing much damage to adjacent cells and axons.

In summary, there is no single animal model that can replicate all of the heterogeneous characteristics of human MS. Each single animal model allows us to study distinct aspects of the disease rather than its entire complexity. Consequently, investigators need to carefully select the animal model to accurately address their specific research question and provide outcomes that result in findings applicable to human MS. Nonetheless, animal models have proven to be highly advantageous in developing novel drugs that are directed toward the process of remyelination.

## 2. The Promotion of Remyelination by the Hh Signaling Pathway

### 2.1. The Contribution of the Hh Signaling Pathway in Myelination and Remyelination

There exist sequential waves of OPC generation in vertebrates, firstly localized in the ventral progenitor domains and later in dorsal regions [53]. Although most generated OPCs differentiate into OLs and contribute to myelination, a small fraction of OPCs remain in a low-proliferative or quiescent state in the adult [54]. Cell-tracing experiments revealed that most adult OPCs are dorsally-derived in the forebrain [55]. After a demyelinating insult, these OPCs undergo activation, proliferation, migration, and differentiation until the formation of new myelin sheaths [56]. As mentioned before, NPCs from in the V-SVZ also can generate OLs after demyelination [8,57,58,59].

The contribution of the Shh signaling pathway in the context of CNS demyelination and remyelination has been addressed by several groups. In addition to its early role in the induction of embryonic OPCs [15], Shh signaling is also implicated in the generation of postnatal OPC populations [60,61]. Exogenous Shh is able to increase the OPC population and premyelinating OLs in the adult forebrain [62]. Genetic cell-fate labeling experiments revealed that neural stem cells (NSCs) in the dorsal V-SVZ respond to Shh and generate OPCs that come to reside in the corpus callosum. These cells persist into adulthood and contribute to remyelination after Cuprizone-induced demyelination [63]. In a model of focal demyelination induced by LPC, the major components of the Hh signaling pathway (including Shh and Smo) were upregulated in the oligodendroglial cells in the area of a lesion. Further gain- and loss-of-function experiments demonstrated that Shh promotes the proliferation and differentiation of OPCs and decreases astrogliosis and macrophage infiltration altogether, leading to the attenuation of the lesion extent during myelin repair [64].

The stimulation of Smo activity is compatible with the positive influence that Shh exerts on remyelination. For instance, the microinjection of SAG, a Smo agonist, into the corpus callosum of LPC-induced demyelinated mouse significantly increased OPC proliferation and enhanced remyelination [65], in accordance with the recent results in Cuprizone-induced demyelinating models [66]. Conversely, GDC-0449 (also referred to as Vismodegib), a specific Smo antagonist, has been reported to repress Gli-mediated transcription in different types of cells [67], significantly aggravating disease severity and increasing the extent of demyelination in the EAE model of demyelination [68].

Recently, another crucial component of the Hh signaling pathway, the type I transmembrane receptor Boc, was identified as a new regulator of myelin formation and repair [69]. During development, Boc forms a Shh receptor with Ptch1 and is necessary for the Shh-mediated proliferation of cerebellar progenitor cells [70]. The Boc-null mutant mice displayed delayed myelination, associated with a reduction in callosal axon diameter. In the context of demyelination induced by LPC injection, Boc was significantly up-regulated in the lesion. During myelin repair, Boc mutants exhibit aberrant OPC differentiation, reminiscent of the phenotypes observed after blockade of the Hh signaling pathway.

### 2.2. Identification of Clobetasol as a Smo agonist for Promoting Remyelination

Although the Hh/Smo signaling pathway possesses an important role in promoting remyelination, the development of related therapeutic strategies has been impeded by the lack of U.S. Food and Drug Administration (FDA)-approved Smo agonists. By using high-throughput screening for cells that express the Smo receptor, four FDA-approved drugs, clobetasol, halcinonide, fluticasone, and flucinonide, were identified as agonists of Smo [71]. These drugs have the capacity to bind Smo, promote the internalization of Smo, cause the activation of Gli factors, and increase the proliferation of neuronal progenitor cells. Meanwhile, several bioactive drugs have also been selected in phenotypic screens for their ability to promote MBP expression in different cell-based assays, including primary OPC cultures [72,73,74], mouse OPC cell lines such as Oli-neuM [75], and epiblast-derived OPCs [76]. These independent drug screens, performed with different libraries and OPC models, support Clobetasol as one of the top-ranking drugs in promoting OPC differentiation and myelin development.

Clobetasol is a member of the glucocorticoid family and is commonly used to treat a number of skin disorders [77]. It is a potential remyelinating agent that has been demonstrated to promote the differentiation of OPCs in vitro, as well as remyelination in vivo [1]. Najm et al. reported that Clobetasol, as a modulator of the glucocorticoid receptor, specifically promotes rapid myelination in organotypic cerebellar slice cultures, as well as in the CNS of postnatal mouse pups [76]. Systematic administration of Clobetasol resulted in a significant increase in newly differentiated OLs and enhanced myelin regeneration in the LPC-induced mouse models of focal demyelination. In an EAE mouse model of chronic progressive MS, an impressive reversal in disease severity was observed when Clobetasol was administrated at the peak of the disease. Furthermore, an assessment of the immune response demonstrated that Clobetasol was able to serve as a robust immunosuppressant in addition to inducing remyelination [76]. In addition, Clobetasol enhanced OL production from human OPCs in vitro [76]. Neuromyelitis optica (NMO) is a CNS disorder that involves inflammation and demyelination of the spinal cord and optic nerve [78]. In a mouse model of NMO produced by an injection of an anti-AQ4 antibody, an intraperitoneal administration of Clobetasol significantly reduced the myelin loss and increased the number of myelinating OLs within the lesions [79]. Recent studies further demonstrated that Clobetasol significantly improved NSC survival and prompted the differentiation of NSCs into neurons and OLs while inhibiting astrocyte differentiation, providing a potentially novel mechanism underlying the therapeutic effect of Clobetasol in CNS-related disease [80].

Altogether, the identified Smo agonist Clobetasol might function in multiple cell types and act via a range of targets to promote myelin repair. Importantly, Clobetasol is able to pass through the blood–brain barrier, raising the exciting possibility that Clobetasol could advance to clinical trials for the currently unavailable chronic progressive phase of MS.

## 3. Negative Regulation of Myelination and Remyelination by the Hh Signaling Pathway

### 3.1. Inhibitory Effect on Myelination by the Hedgehog Signaling

In the transgenic mice that ectopically expressed Shh in the dorsal neural tube, spinal precursor cells were arrested in an undifferentiated state and exhibited elevated levels of proliferation [81]. Recently, our team discovered a stage-specific activity of Hh signaling in OL development and showed that persistent activation of Smo in OPCs inhibited their differentiation [82]. Thus, Smo-mediated Hh signaling appeared to robustly promote NPC or OPC proliferation and resulted in the inhibition of OPC differentiation and subsequent myelination during early developmental stages. This observation is in agreement with the blockade of myelin development by the Smo agonist, SAG [65]. Moreover, the fact that appropriate myelination during development requires down-regulation of Hh is consistent with the thin corpus callosum observed in patients with Gorlin syndrome [83]. This syndrome is associated with a mutation in the Hh receptor, Ptch1, that blocks the repression of Smo activity, allowing for the increased activation of Hh signaling. In summary, Smo-mediated Hh signaling has an apparent inhibitory effect on OPC differentiation and developmental myelination.

### 3.2. Down-Regulation of Gli1 during Myelination and Remyelination

In fact, Gli1, originally considered to be a reliable readout of Hh signaling activity, has proven to be detrimental during myelination and remyelination. During development, the genetic ablation of Gli1 in NPCs appeared to lead to precocious myelination [11]. Specifically, the inhibition of Gli1 through specific-inhibitor GANT61 in human iPSCs-derived neural stem cells (NSCs) resulted in the increased generation of OPCs. These GANT61-induced OPCs are more migratory, in agreement with the single-cell RNA sequencing that show up-regulated cytoskeletal reorganization pathways. The differentiated OLs were proven to be functional and able to generate compact myelin both in vitro and in vivo [84]. Thus, the inhibition of Gli1 in NSCs facilitates OPC generation and OL maturation during development.

In addition, the negative regulation of myelin regeneration by Gli1 was also reported recently. During the demyelination and remyelination processes, the expression of Gli1 appeared to be variable depending on the animal models that were used. When demyelination was induced in the corpus callosum by an injection of LPC, a relatively moderate transcription of Gli1 was seen in OPCs within the lesions [64]. In the EAE model, Gli1 transcription was up-regulated in OPCs and neurons immediately before EAE onset but down-regulated while the demyelination stage [85]. Concerning the Cuprizone model, it was noted that little to no up-regulation of Gli1 was observed in the demyelinated corpus callosum, primarily in the reactive astrocytes [66]. Importantly, fate-mapping experiments following Cuprizone-induced demyelination showed that a subset of SVZ-derived Gli1-expressing NPCs down-regulated Gli1 expression upon arrival to the lesion site [11]. Moreover, the inhibition of Gli1 expression in the Cuprizone model was found to amplify the recruitment of NPCs, promote the migration of OPCs to the demyelinated axons, and enhance remyelination [84,86]. Concomitantly, the pharmacological inhibition of Gli1 activity directly or indirectly improved the functional outcomes in the EAE model by promoting remyelination and neuroprotection in the spinal cord [11]. It is noteworthy that this role appeared to be specific to the inhibition of Gli1 independent of the canonical Hh signaling, based on the observation that remyelination was unaltered when the canonical Hh signaling was inhibited.

## 4. The Complex Involvement of Canonical and Non-Canonical Hedgehog Signaling Pathways in Remyelination

As described above, the use of several small molecules able to activate Smo-mediated Hh signaling or to block its key effector Gli1 led to improved remyelination in several animal models (Table 1), which might reflect not only the potential involvement of canonical and non-canonical Hh signaling pathways, but also the targeting of different cell types (NSCs or resident OPCs), and probably the differential activation of local inflammatory cells.

### 4.1. The Promotion of Remyelination via the Non-Canonical Pathway

As mentioned above, Smo is able to transduce Hh signaling via both canonical and non-canonical pathways [33,34,35,36,87,88]. In agreement with the findings that Gli1 inhibition by GANT61 improves remyelination [11], the non-canonical Smo agonist GSA-10 has been recently reported to promote remyelination (Table 1) [87]. GSA-10 was first identified through a Smo pharmacophore-based screen [89,90], and it belongs to a new family of Smo agonists that activate the non-canonical pathways associated with Gli1 inhibition [91]. In the Oli-neuM cell line, GSA-10 was a potent activator of OPC differentiation. Upon demyelination induced by LPC, it prompted the OPC recruitment toward the lesion area without enhancing their proliferation. Notably, GSA-10 displayed the ability to promote OL maturation up to the stage of engaging artificial axons [87]. In conclusion, non-canonical Hh signaling is able to promote remyelination until axon engagement, representing a novel potential therapeutic target. Thus, together with the remyelinating effects described for the other small molecules binding Smo, the conspicuous remyelinating effects of GSA-10 support the idea that different Smo agonists can activate distinct signaling pathways presumably by activating Smo at different sites [36,89,90]. Interestingly, Smo activation by GSA-10 led to Gli2 upregulation, which is consistent with the recent report that ablation of Gli1 increased the expression of Gli2 in NPCs following Cuprizone-induced demyelination [86]. In the same line, Sox17 has also been found to induce OL regeneration in demyelinated areas through an increase in Shh/Smo/Gli2 activity [92], further supporting the importance of Gli2 upregulation for the differentiation program under Gli1 downregulation.

### 4.2. Hh Signaling Modulation Controls Local Inflammatory Cells

Under repairing conditions, inflammatory cells in the affected regions, including astrocytes and microglia, are endowed with beneficial or deleterious properties, promoting or impairing the endogenous capacity of OPCs to induce spontaneous remyelination after myelin loss [3,93]. Therefore, astrocytes and microglia are becoming additional targets for assessing remyelinating properties afforded by small molecules. The observation that SAG was able to promote OPC differentiation in the context of demyelinated lesions was unexpected given its ability to also promote OPC proliferation [65,66]. During spontaneous remyelination occurring after LPC-induced demyelination, the Smo receptor is up-regulated in OLs and microglia but at a reduced level in astrocytes. Upon demyelination, SAG might promote the differentiation of OPCs indirectly by influencing microglia, inducing the expression of anti-inflammatory markers. This potential mechanism is supported by the previous report that microglia were shifted to an anti-inflammatory phenotype that could direct OL differentiation during remyelination [94]. Consistently, the conditional removal of Smo from microglia resulted in a dramatic decline of differentiated OLs, suggesting that Smo is cell-autonomously required for the response of microglia to a demyelinating event and that the pro-differentiating activity of SAG is related to its influence on microglia. Although GFAP expression did not appear to be regulated by SAG, the selective up-regulation of Smo in astrocytes in the LPC models also raises the possibility that its pro-differentiating activity might be mediated by specific subsets of astrocytes.

## 5. Conclusions

Given that the process of remyelination is extensive and complicated, boosting remyelination is a current challenge in the field of CNS demyelinated disease. Current approaches for discovering regenerative therapies in MS are mostly based on the assumption of enhancing the differentiation of resident OPCs to OLs. However, a recent single-cell analysis of the white matter in MS patients identified oligodendrogial heterogeneity in MS and proposed that strategies to restore healthy OL heterogeneity should be a major focus in the future treatment of MS [12,13]. Therefore, in addition to the generation of new OLs, spared OLs during the demyelinating process might also be the target of remyelinating therapies. The data accumulated over the years regarding the remyelinating properties of Hh signaling modulators have uncovered multiple roles that target several steps of the regeneration process as well as several cell types. Both canonical and non-canonical Smo-mediated activities likely contribute to tissue regeneration in an intricate manner opening the way to new perspectives in the therapeutic of CNS demyelination diseases.

## Figures and Tables

**Figure 1 cells-11-02260-f001:**
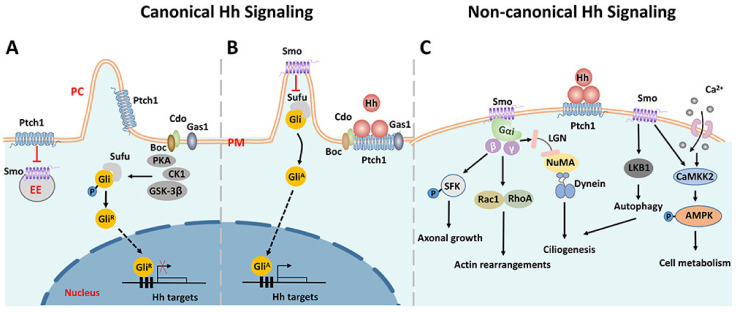
Schematic diagram of Smo-driven canonical and non-canonical Hh signaling pathways. (**A**) In the canonical Hh signaling, when Hh ligands are absent, Ptch1 exerts a repressive activity on Smo. Gli factors are sequestered by Sufu and phosphorylated by PKA, CK1, and GSK-3β, and then cleaved by post-translational proteolytic processing. The truncated Gli factors translocate into the nucleus and repress the transcription of downstream genes. (**B**) Upon Hh binding to Ptch1, the repression of Smo is relieved and Smo accumulates in the primary cilium, inhibiting the sequestration and phosphorylation of Gli factors. The full-length Gli factors then translocate into the nucleus and ultimately activate the transcription of downstream genes. (**C**) Representative non-canonical Hh signaling pathways dependent on Smo. Smo can activate several downstream effectors involved in various biological processes. PM: plasma membrane. PC: primary cilium. EE: early endosome. ER: endoplasmic reticulum.

**Table 1 cells-11-02260-t001:** Summary of the effects of reported pro-remyelinating Hh modulators.

Name	Hh Modulation	Pro-remyelination Effects	References
SAG 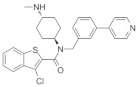	Smo agonist, activates canonical and non-canonical Hh pathway	Stimulates OPC proliferation and differentiation during myelin repair in LPC and cuprizone models	[65,66]
GSA-10 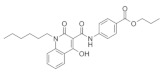	Smo agonist; activates non-canonical Hh pathway	Up-regulates Gli2 expression; Promotes the recruitment and differentiation of OPCs in LPC models	[87]
Clobetasol 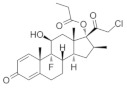	Smo agonist	Up-regulates MBP expression in Oli-neuM cell line; Increases NSC viability and promotes NSC and OPC differentiation; Promotes OL maturation up to axon engagement; Reverses the disease severity in EAE models; Promotes remyelination in NMO models	[75,76,79,80]
Halcinonide 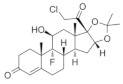	Smo agonist	Up-regulates MBP expression in Oli-neuM cell line	[75]
Gant61 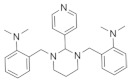	Gli1 antagonist	Increases the generation of OPCs from iPSC-derived NSCs; Enhances the migration of OPCs; Promotes myelin repair in Cuprizone models	[11,84]

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
