# Peer review of "Hedgehog Signaling in CNS Remyelination"

_cells, 2022, doi:10.3390/cells11142260_

Round 1
Reviewer 1 Report
Fang and colleagues have written a review entitled “Hedgehog signaling in CNS remyelination”. The review discusses the significance of Hh signalling in the process of remyelination, with a focus on the current controversies and potential issues. However, I find the organisation of the sections not easy to interpret. With modifications and editing to the text, it could be improved. I have the following specific comments:
1. The introduction of the manuscript suggests that the review will discuss Hh signalling with a focus on the demyelinating disease Multiple Sclerosis. It also nicely introduces the various animal models of MS in section 1.3. However, the manuscript does not retain this focus.
The information on Hh signalling is not broken down into these different models. The text is a mixture of the models together and interchanges between them. In addition, other models are discussed which were not introduced in the section, such the mouse model of neuromyelitis optica or cerebellar slice cultures, and mostly in vitro experiments on OL. I am therefore unsure of the relevance of a dedicated section discussing the different MS models, but not discussing the role of Hh therapy in these models. I would be curious to know, could the variable results seen with Hh targeting be related to the different mechanisms of oligodendrocyte cell death/demyelination? For example, in models with an intact immune system (EAE) versus one with less involvement (Cuprizone). Or are there better results in animal models in which the BBB is disrupted due to better drug access? It would have been nice to have the existing in vivo data reviewed in the paper broken down into the different models. This could be then highlighted within the included table. The authors quite rightly state that the etiology of MS is complex, therefore understanding what Hh agonists do or do not do in the different in vivo animal models would offer insights into its therapeutic efficacy.
2. The three Hh proteins Shh, Ihh and Dhh are introduced but what is their individual relevance in the control of OPC development?
3. Section 3 which describes the development of smo agonists and their issues making it the clinic, comes after section 2 which already details the use of smo agonists in remyelination. It would perhaps make more sense to describe the agonists first, then describe their potential positive roles in remyelination. Section 2 and 3 could therefore be merged and better organised.
4. Since the smo agonist Clobetasol might act on multiple cell types with a range of targets in the CNS, could there be potential issues with off-target side-effects? Authors go on to discuss astrocyte and microglia as Hh signalling targets. Could this make agonists such as these harder to translate to the clinic?
5. Although the non-canonical Hh signalling pathway appears more beneficial for controlling remyelination, what is the specificity of smo agonists to target only this pathway and not bind smo via the canonical pathway? Is there any evidence of their effects in vivo?
6. There are statements that I think require referencing
-“Various demyelinating models were used to show the relevance of the Hh signaling in myelin regeneration”.
- “Interestingly, these independent drug screens, performed with different drug libraries and OPC models, support clobetasol as one of the top ranking drugs in promoting OPC differentiation and myelin development”.
These statements needs to be qualified with references. I realise the authors describe a selection of these models either before or after, but references should be placed after these statements. Is clobetasol top ranking in their opinion based on their interpretation of the screen data or is this reference based?
Minor criticisms:
7. In general, the English content of the manuscript is acceptable, however it requires to be strengthened with proof reading e.g.
-Within the introduction “major causes for axonal demyelination is chronic inflammatory in the CNS” should instead read chronic inflammation.
-“Remyelination involving the formation of new myelin sheath” should read formation of new myelin sheaths.
-“Three Hh proteins have been discovered in vertebrates, including Sonic Hedgehog (Shh), Desert Hedgehog (Ihh) and Indian Hedgehog (Dhh)” abbreviations within brackets are wrong.
Reviewer 2 Report
Comment to the author
The authors make a clear and concise review of the state of art on the role of Hh signalling stimulation during adult remyelination.
I appreciate their effort although I suggest changing phrases containing the words "contradictory" and "Conflictual" that can be missinterpreted by reader, as in general the review gives a positive view of the beneficial effects that promyelinating agents targeting Hh signalling pathway. However, some phrases indicate the existence of a "contradictory" or "conflictual" data that instead are nor contradictory or conflictual, just they have been obtained in different experimental models and data, as suggested by the author can be reinterpreted. Therefore I suggest the authors to rewrite these phrases to give a more appropriate vision of the current state of art, that essentially show that Smo-mediated signalling is activated at two stages of the process of remyelination: the first time during NPC to OPC reactivation via a canonical pathway leading to proliferation and in a second step, after Gli1 downregulation, at the stage of OPC to OL differentiation (see further explanation below) via a non-canonical pathway
Below my comments:
Major points:
-Abstract: lines 22 -24 “This review summarises the current “controversy “concerning the role of the Hh signalling pathway in the process of remyelination, attempting to address the plausible mechanism underlying this controversy and the potential issues in targeting the Hh signalling pathway as a potential demyelinating therapy.”
I do not see conflicts or controversy in the current data, just old data must be revised according to the new findings obtained in different models and using drug treatment targeting the Hh pathway during remyelination. Therefore, I found the word “controversy” in the text not adequate to explain authors’ point of view.
My suggest to change the phrase in lines 22 -24 as follow to give the correct message: This review summarises the current data concerning the role of the Hh signalling pathway in the process of remyelination, we will highlight the open issues that have to be clarified prior to bring molecules targeting the Hh signalling to the demyelination therapy."
-Lines 147-149 please corect the words “conflictual experimental results ” as it is misleading. Data are not conflictual
-Lines 151-152: Delete phrase see comment below
-Lines 328-332: Delete phrase see reasons below
Essentially it is now clear that the process of remyelination passes through phases of Hh signaling activation via canonical pathway (Ferent et al., 2014, Samanta et al., 2015), and through stages of Smoothened (Smo) activation via a non-canonical pathway (Samanta et al., 2015, Del Giovane et al., 2022). Specifically, all data, correctly cited by the author, show that NPC reactivation and proliferation requires canonical SMO/Gli1 signalling activation (exemplified by SAG-type of stimuli). However, upon NPC-fating to OPC, that is signed by Gli1 downregulation (Samanta et al., 2015), a non-canonical pathway is activated that leads to Gli2/AMPK signaling paralleled by Gli1 downregulation that leads to OPC differentiation (Del Giovane et al., 2022). Therefore, available data, are not conflictual but has to be revised accordingly, as suggested by the authors. Two different signals activate Smo during remyelination process: one stimulates via Shh the Patch/Smo/ Gli1 signalling (proliferation pathway) leading to NPC proliferation and fating to OPC, the second signal, activated by other endogenous signals, possibly cholesterol and oxysterol (Lucchetti et al., 2017) activates the Smo/Gli2/AMPK/MBP pathway. Therefore, Smo modulators must be characterised for their effects on both NPC and OPC before to be proposed for clinical therapy for remyelination.
Minor points
Figure 1.
Figure 1A and 1B: please indicate in the scheme also Gli factors.
Figure 1C: Ptch1 indication is missing in the scheme. Figure Background white would be better as it is better copied
Ref 73: The first name of reference is miss typed. It should be : Del Giovane A, Russo M, Tirou L, Faure H, Ruat M, Balestri S, Sposato C, Basoli F, Rainer A, Kassoussi A, Traiffort E, Ragnini-Wilson A. Smoothened/AMP-Activated Protein Kinase Signaling in Oligodendroglial Cell Maturation. Front Cell Neurosci. 2022 Jan 10;15:801704. doi: 10.3389/fncel.2021.801704.
Reviewer 3 Report
This review summarizes the known data about the role of Hedgehog signaling in the myelination process. The review has a nice flow and it is easy to follow, but I feel some sections need a bit more information that would make it complete.
The description of signal transduction is nicely presented for the membrane part of the signaling pathway, but nothing is written about the cytoplasmatic components and the regulation of GLI proteins by kinases. Also, non-canonical HH signaling has two branches: one is presented here, and it includes redirection of the signal from SMO to non-pathway proteins. Another is the activation of GLI proteins by other signaling pathways independently of the membrane components. This branch should at least be mentioned when describing the non-canonical signaling.
„...Gli2 and Gli3 are thought to act as negative or positive regulators of the canonical Hh signaling pathway...“ written like this, it looks like Gli2 is negative while GLI3 is positive, while the opposite is true. I suggest changing the order of negative and positive to make it clearer
I would recommend adding some data on the effect of GLI2 and GLI3 on myelination/demyelination process. This might explain the contradictory results seen with GLI1, as the whole GLI code may be involved in the process. For the review to be complete I would like to see some comment on this as well.There are some publications investigating the role of GLI2 and GLI3 in this process, and one is very fresh, published this year, and it may not have been published at the time of submission.
